# Mapping Research Trends of Adapted Sport from 2001 to 2020: A Bibliometric Analysis

**DOI:** 10.3390/ijerph191912644

**Published:** 2022-10-03

**Authors:** Tao Liu, Nicole Wassell, John Liu, Meiqi Zhang

**Affiliations:** 1School of Kinesiology, Beijing Sport University, Beijing 100084, China; 2Department of Physical Education and Health Education, Springfield College, Springfield, MA 01109, USA

**Keywords:** adapted sport, Paralympics, Special Olympics, intellectual disability, sport injury

## Abstract

Objectives: To identify the research landscape in terms of keywords, annual outputs, journals, countries, and institutions and explore the hot topics and prospects regarding adapted sport research. Materials and methods: Publications designated as “article” on adapted sport retrieved from the Web of Science Core Collection. VOSviewer 1.6.11, Citespace, and Bibliometrix in R Studio were applied for the bibliometric analyses. Results: A total of 1887 articles were identified. Over the past two decades, athletic performance, sociology/psychology, and rehabilitation were extensively investigated. Basketball, soccer, and swimming were the three most focused adapted sports. Researchers showed a growing interest in submitting their studies to sport science, rehabilitation, and sociological journals. Adapted sport research was more common in developed countries and regions. The UK contributed most publications accounting for about 20% of the total publications. Conclusions: With the growth of publications concerning adapted sport, the bibliometric analysis presented an overview of collaboration, trends, and hotspots in the field.

## 1. Introduction

Participation in sports has shown many benefits for the mental and physical health of individuals with disabilities [1,2]. Adapted sports make use of adapted equipment or rules to make sport accessible to individuals with disabilities [1,3]. A wide range of sports have been adapted to be played by individuals with varied disabilities. According to the largest adapted sports event, the Paralympic games, the first Paralympics in 1960 hosted about 400 athletes from 23 countries/regions participating in eight sports; while at the 2020 Summer Paralympics in Tokyo, more than 4400 athletes representing 162 countries participated in 22 sports [4]. In just over half a century, adapted sports have expanded and globalized.

The thriving of adapted sport is also reflected in scientific research. The number of literature studies grows significantly. It is necessary to analyze the literature by taking into consideration the development of research topics, hotspots, journals, authors, institutions, countries, and collaborations. The bibliometric analysis adopts statistical methods to demonstrate knowledge structures and dynamic evolutions of a specific research field. As a quantitative approach, the bibliometric analysis can provide a comprehensive view of a research concentration/topic/subject. In the field of adapted sport literature, Khoo et al. [2] performed a bibliometric analysis exploring the distribution of the top 50 most cited articles in terms of research categories, authors, institutions, and journals. An earlier study performed a documentary analysis of research priorities in adapted sport between 1986 and 1996, which involved 436 articles [5]. In the present study, we aimed at investigating adapted sport literature by evaluating the impact of countries, institutions, authors, journals, and topics. The findings of this study may provide all-round insights and suggestions about the global trends in adapted sport research.

## 2. Materials and Methods

### 2.1. Source of Data and Search Strategy

Published papers were retrieved from a topic search in the Web of Science Core Collection (WosCC; Clarivate Analytics, Philadelphia, PA, USA) in October 2021. The following search strategy was used: TS (topic search) = (Paralympic* OR adapt* + sport* OR disab* + sport* OR parasport* OR special + olympic* OR unified + sport* OR inclusive + sport* OR deaflympic* OR sport* + for + disabled OR sport* + for + handicapped OR sport* + for + the + impaired OR sport* + for + physical + disabilit* OR sport* + for + sensory + disabilit* OR sport* + for + special + needs OR blind* + sport* or deaf + sport* OR wheelchair + sport* OR parasport*) and time span = 2001–2020.

### 2.2. Inclusion Criteria

The inclusion criteria are shown in Figure 1. Only studies designated as “article” were considered. Any other document types such as meeting abstracts and letters were excluded. Full records and cited references of the publications were imported from the WoSCC. Only literature published in English was included. If a publication was attributed to more than one country/region or institution, the correspondence authors’ information is considered. Finally, a total of 1887 records published during the period 2000–2019 met the inclusion criteria.

### 2.3. Analytical Methods

VOSviewer 1.6.11 [6], Bibliometrix on R studio [7], and CiteSpace [8] were used to identify and visualize the critical information of the publications. VOSviewer was used to develop the social network maps for co-occurrence keywords. The 20 years were divided into four five-year periods, including 2001–2005, 2006–2010, 2011–2015, and 2016–2020, to depict the time-dependent evolution of literature. GraphPad Prism 8.0.0 [9] was applied to perform the time trend of annual publication outputs.

## 3. Results

### 3.1. Trends in Publications on Adapted Sport

From 2001 to 2020, a total of 1887 papers on adapted sport were recorded in the WoSCC database based on the above retrieval strategy. Figure 2 shows the publication output of adapted sport research from 2001 to 2020. The number of annual publications continuously grew from 16 in 2001 to 241 in 2020. Most research was published in 2018 (*n* = 249 publications, 13.2%).

According to the correspondence author’s country, over the past two decades, a total of 66 countries/regions had contributed to adapted sport research. In the period of 2001–2005, only 20 countries contributed to adapted sport research; while during 2016–2020, researchers from 62 countries were devoted to such research. The top ten countries/regions (by corresponding author’s country) were composed of five European countries, two North American countries, a South American country, an Asian country, and an Oceanian country. As Figure 3 shows, the United Kingdom (UK) was the most contributive country (*n* = 398 publications) accounting for 21% of the total publications, followed by The United States of America (USA, *n* = 346), Canada (*n* = 165), Australia (*n* = 122), Brazil (*n* = 103), Spain (*n* = 71), Italy (*n* = 65), Japan (*n* = 55), Netherlands (*n* = 53), and Germany (*n* = 47). Table 1 has held the top ten countries/regions over the years from 2001 to 2020 in different periods of time. It is noteworthy that the UK has made significant progress from 38 publications during 2006–2010 to 165 during 2011–2015 and maintained the top position in the following five years. A co-authorship analysis was compiled to reveal international collaborations, as shown in Figure 4.

A total of 1849 institutions participated in adapted sport research from 2001 to 2020. The top ten institutions were: Loughborough University (*n* = 103 articles, UK), University of British Columbia (*n* = 48, Canada), University of Pittsburgh (*n* = 45, USA) and University of Sunshine Coast (*n* = 45, Australia), University of Groningen (*n* = 42, Netherlands), Catholic University of Leuven (*n* = 40, Belgium), University of Queensland (*n* = 38, Australia), German Sport University Cologne (*n* = 36, Germany) and Harvard Medical School (*n* = 36, USA), and Vrije University Amsterdam (*n* = 35, Belgium). The top ten institutions are shown in Table 2.

From 2001 to 2020, research relating to adapted sport has been published in 612 scholarly journals. The top ten journals for the 20 years and each five-years period are presented in Table 3. Over the past 20 years, *Adapted Physical Activity Quarterly* (*n* = 69 articles) published the most adapted sport research, the following were *British Journal of Sports Medicine* (*n* = 51), *Disability and Rehabilitation* (*n* = 37), *International Journal of Sports Physiology and Performance* (*n* = 33), *Sport in Society* (*n* = 32), *Journal of Sports Sciences* (*n* = 30), *Clinical Journal of Sport Medicine* (*n* = 28), *Proceedings of the Institution of Civil Engineers-Civil Engineering* (*n* = 28), *PM&R* (*n* = 22), and *Disability & Society* (*n* = 21). Half of the journals were related to sport science while the remaining included sociology, rehabilitation, and engineering journals.

Analysis by category was performed (Figure 5): sport sciences (*n* = 653 publications), rehabilitation (*n* = 368), hospitality leisure sport tourism (*n* = 292), physiology (*n* = 101), applied psychology (*n* = 87), sociology (*n* = 84), educational research (*n* = 73), public environmental occupational health (*n* = 73), special education (*n* = 67), and orthopedics (*n* = 58).

A total of 5328 authors contributed to the research of adapted sport. The publication counts in Table 4 reveals that Goosey-Tolfrey V.L. (*n* = 44 publications) was the most productive author over the past two decades, followed by Burkett B. (*n* = 40), Van De Vliet P. (*n* = 27), Derman W. (*n* = 26), Webborn N. (*n* = 22), Mason B.S. (*n* = 20), Smith B. (*n* = 20), Reina R. (*n* = 19), Howe P.D. (*n* = 17), and Tweedy S.M. (*n* = 17). In terms of co-cited counts, Webborn N. (*n* = 289 citations) ranked first as the most co-cited author, followed by Makhov A.S. (*n* = 276), Van De Vliet P. (*n* = 276), Tweedy S.M. (*n* = 273), Derman W. (*n* = 248), Howe P.D. (*n* = 236), Goosey-Tolfrey V.L. (*n* = 216), Stomphorst J. (*n* = 207), Burkett B. (*n* = 205), and Vanlandewijck Y.C. (*n* = 205).

### 3.2. Research Hotspots and Keyword Detections

The ten keywords with the highest occurrence frequencies were: performance (*n* = 220), people (*n* = 174), exercise (*n* = 165), sport (*n* = 152), children (*n* = 144), physical-activity (*n* = 128), disability (*n* = 122), individuals (*n* = 119), participation (*n* = 107), and spinal-cord-injury (*n* = 100).

A clustering analysis of keywords was performed with VOSviewer (Figure 6). With a threshold of ≥20 occurrences, 126 keywords were selected, and four clusters emerged by the co-occurrence clustering analysis. The four clusters represented by different colors are as follows: (1) sport performance in red; (2) sociological, cultural, and educational research in green; (3) special Olympics in blue; and (4) health promotion in yellow. Figure 7 shows the analysis of the keywords of adapted sport publications over the past two decades. The publication year is demonstrated as a gradient color change. Figure 8 shows the cluster analysis of keywords over the most recent five years (i.e., 2016–2020). With a threshold of ≥10 occurrences, 109 keywords were identified and were classified into five clusters: Sport performance; sociological, cultural, and educational research; sport injury; adapted sport in youth; health promotion.

Figure 9 presents the keywords with strongest citation bursts in adapted sport research. The most recent burst keywords were configuration, supercrip, and physical disability.

The most focused on sports in adapted sport research were further analyzed by Bibliometrix (Figure 10). Over the past 20 years, the ten sports that received the most attention are: basketball (*n* = 32), soccer (*n* = 26), swimming (*n* = 22), baseball (*n* = 14), football (*n* = 13), racing (*n* = 10), rugby (*n* = 8), sprint (*n* = 8), tennis (*n* = 6), and combat (*n* = 6).

## 4. Discussion

Bibliometric data provides a comprehensive understanding of the impact of publications. Only one bibliometric analysis related to adapted sport [2] was found by the literature retrieval, which identified the 50 most cited publications between 1993 and 2014. The study provided valuable information for the present research. Based on this, the bibliometric analysis included more and recent publications to investigate the characteristics of publications and provide insights into research trends and hotspots.

Annual outputs can reflect scientific inputs and socioeconomic supports. The annual outputs of adapted sport research have kept increasing over the past 20 years, especially since 2010, there has been a surge in annual outputs. The tendency of publications indicated the ongoing scientific investment into adapted sport. However, given the increased sport participation in people with disabilities, the research productivity should be matched to the increased actual demands. The authorities and relevant social organizations may highlight and invest in research as much as developing events and games.

More adapted sport research came from high-income countries, western European and North American countries. Greater sport participation, together with relatively sufficient support, assured the long-term research of adapted sports in the countries and consolidated the leading global positions in athletics. The growing attention in adapted sport research could also be attributed to playing hosts of relative events. Brazil had observed a surge and ranked seventh among the top ten countries in 2006–2010 and fifth over the 20 years. As the only developing country of the top ten countries, preparing for, and holding the 2016 Rio Paralympic Games should be an inseparable factor. A similar trend is also observed in China.

Furthermore, although over sixty countries have been involved in adapted sport research, publication output is distributed unevenly between institutions. According to the co-authorship map, the overall links between countries were weak, indicating less collaboration among countries and less willingness to collaborate except for the network consisting of numerous institutions in Canada and Australia. In general, institution collaboration should be further strengthened.

Throughout the frequency distribution of 612 journals, the top ten journals in the field were the main sources of publications. The *Adapted Physical Activity Quarterly* ranked first over the 20 years, reflecting an extraordinary impact on the research field. By comparison, in the past 20 years, more researchers have placed attention on sport science than sociological studies. This may be attributed to the limited number of yearly publications in sociological/humanin journals, but this may be also attributed to the fact that the increased competition in adapted sport events promoted sport science research. In short, sport science-related studies may have more choices in submitting, but more attention should be given to sociological/humanin related studies [10,11,12].

Co-occurrence keywords were identified to explore the preference of studying subjects and the tendency of research frontiers. Although most of the frequent keywords only provided unspecific information on the publication trends, two valuable insights were still able to be found as follows. First, athletic performance, sociology/psychology, and rehabilitation/injury prevention attracted more attention during the past two decades. The finding was like the previous bibliometric analysis, which reported that sociological and psychological aspects and athletic effects occupied more proportion of publications [13].

The analyses provided information on who is standing at the frontier of this research area:

Sport injury and rehabilitation in adapted sport is an emerging hotspot in the most recent years. Such topics involved the risk factors and incidence of injuries [14,15], shoulder pain in wheelchair athletes [16,17], acute and chronic injury [18,19], and related prevention and intervention efforts [20,21], etc.

Psychology in adaptive sports has received increased attention over the years. Researchers were intensely concerned mental health of the athletes throughout their athletic stages of development: (1) the therapeutic potential of participation in sport for mental/emotional health [3]; (2) motivation, barriers, and facilitators that may impact sport participation and athletic performance [3,22,23]; (3) the mental/emotional impact of retirement from sport [24,25].

As for the sociocultural study, except for conventional topics such as gender [26,27,28] and race [29], supercrip [30] was the most discussed or critiqued in the field of study [31,32].

Overweight/obesity and dental problems have been severe health concerns for athletes with intellectual disabilities. Researchers exposed the fact that participation in sports would not offset the risk of overweight/obesity in this population; moreover, female athletes with intellectual disabilities tended to have a higher incidence of overweight/obesity than males across ages and regions [33,34,35]. As for dental problems, an amount of epidemiological research investigated the prevalence in worldwide athletes with intellectual disabilities [36,37,38]; some public and research projects had been implemented (e.g., Special Olympics Special Smiles and Special Smiles) but the effect seemed to be limited, suggesting that continuous efforts for preventive and restorative oral health are needed in this population [39,40].

None of the top sports in the last 20 years have discussed winter sports, which suggested fewer concerns and underdeveloped research. For people with physical disabilities, ice/snow sports provide the opportunity for them to move with agility and speed not possible under their own power on land [1]. Therefore, winter sport is an essential component of adapted sports, and adapted sports researchers should pay more attention to it.

Some potential limitations were present in this study. First, the study only focused on literature indexed from WoSCC because of the most authoritative and credible publication information being provided. Inevitably, some useful information might have been missed from other databases (e.g., PubMed and Embase). Second, due to the large number of included articles, only the titles and abstracts of each publication were identified. Thus, football and running failed to classify, which might lead to potential inaccuracy of included key terms compared with manual annotation. Third, there might be some biases caused by non-language restrictions in the search strategy. To achieve a broader range of evaluation, there was no language restriction in the study. While English publications were considered to have a higher impact than non-English publications, which might result in language bias. All the aforementioned factors might affect the accuracy of the bibliometric analysis.

## 5. Conclusions

The present research served as the first attempt to assess the adapted sports literature by a quantitative approach. The analysis provides a comprehensive overview of adapted sports research between 2001 and 2020. The annual publication outputs continually increased between 2001 and 2020. The Adapted Physical Activity Quarterly, British Journal of Sports Medicine, and Disability and Rehabilitation ranked top three productive journals over the past two decades. The UK, the USA, and Canada were the three major contributors to adapted sports research. Sports injury/rehabilitation, psychological, and sociocultural topics were the main forces of adapted sports research.

This analysis also depicted research trends of adapted sports research: (1) sports injury and rehabilitation in adapted sports have grown rapidly in recent years; (2) the therapeutic potential of participation in sports, factors that may impact sport participation and performance, and mental/emotional impact after retirement were highly concerned in psychological research; (3) Supercrip was a newly discussed sociocultural hotspot in adapted sports research; (4) obesity and dental issues had been serious health problems in athletes with intellectual disabilities, preventative and restorative strategies should be explored; (5) winter sports should be an essential part of adapted sports but still less concerned.

## Figures and Tables

**Figure 1 ijerph-19-12644-f001:**
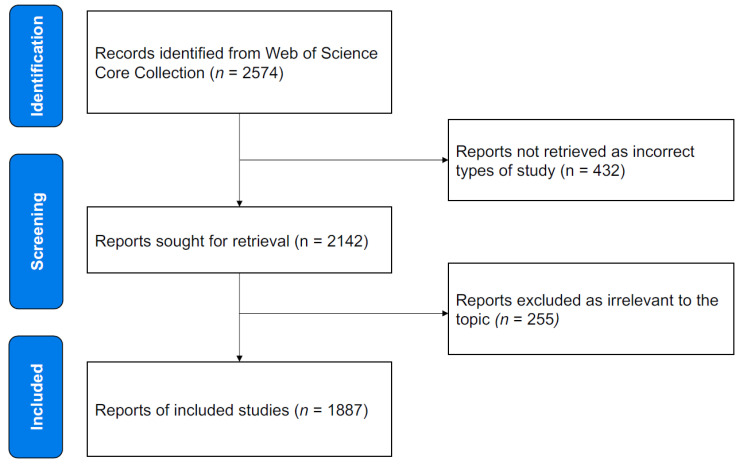
Flow chart of adapted sport studies inclusion.

**Figure 2 ijerph-19-12644-f002:**
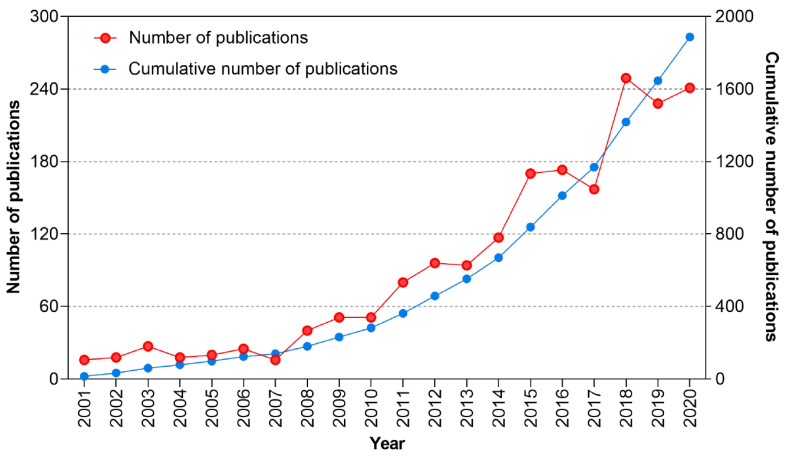
Annual number of publications and cumulative number of publications by year.

**Figure 3 ijerph-19-12644-f003:**
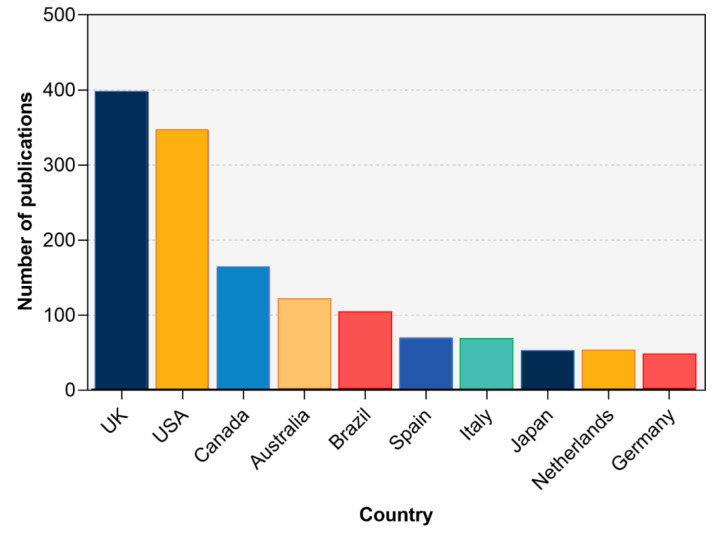
Top 10 productive countries on adapted sport research.

**Figure 4 ijerph-19-12644-f004:**
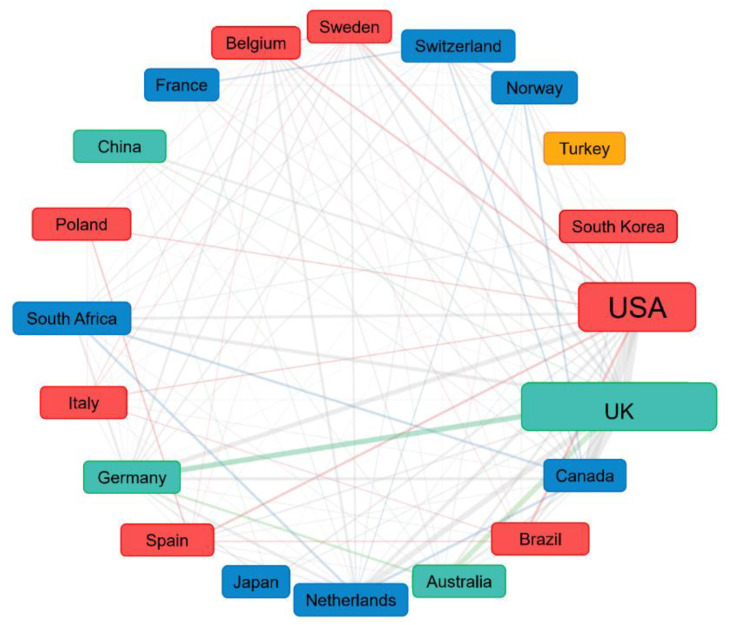
International collaborations on adapted sport research.

**Figure 5 ijerph-19-12644-f005:**
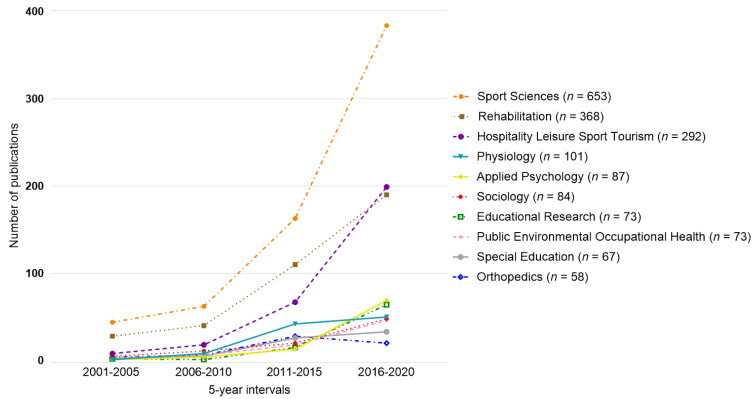
Major changes in the top 10 categories of adapted sport research.

**Figure 6 ijerph-19-12644-f006:**
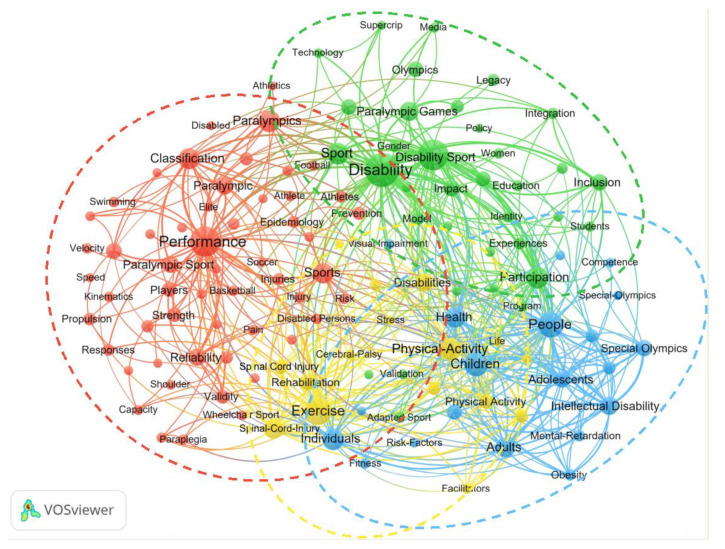
Keywords co-occurrence analysis of adapted sport publications between 2001 and 2020.

**Figure 7 ijerph-19-12644-f007:**
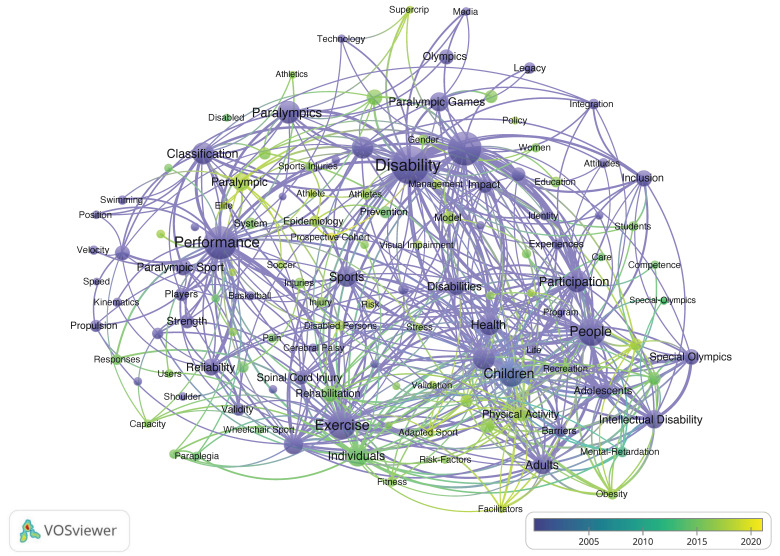
Keywords co-occurrence analysis of adapted sport publications between 2001 and 2020 with timeline.

**Figure 8 ijerph-19-12644-f008:**
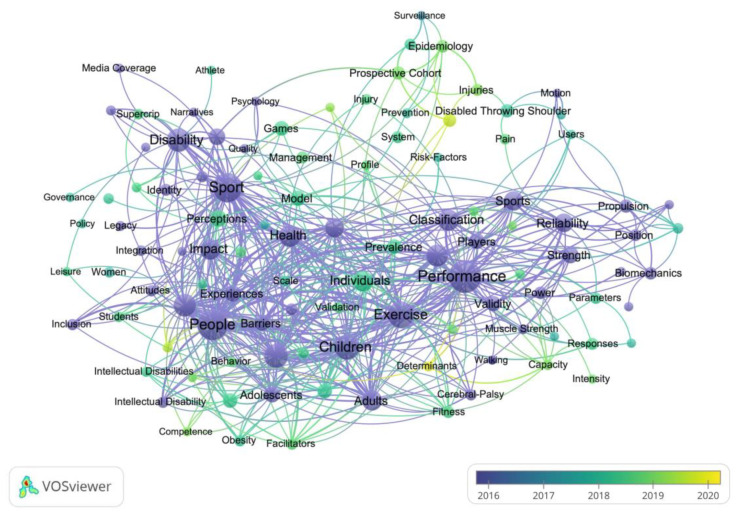
Keywords co-occurrence analysis of adapted sport publications between 2016 and 2020 with timeline.

**Figure 9 ijerph-19-12644-f009:**
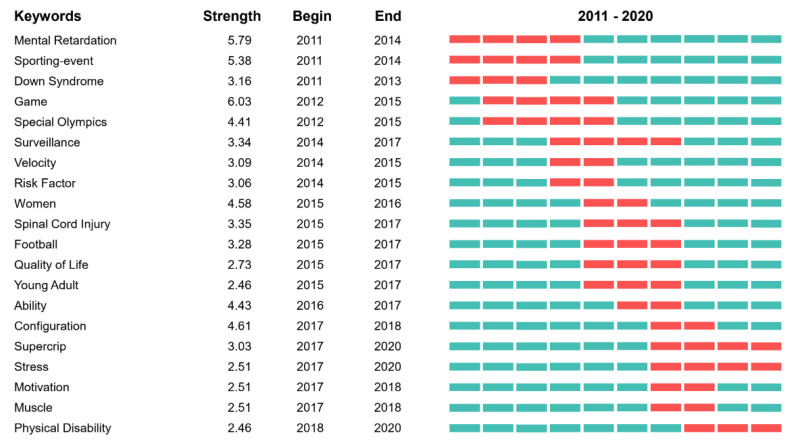
Keywords with the strongest citation bursts of publications on adapted sport.

**Figure 10 ijerph-19-12644-f010:**
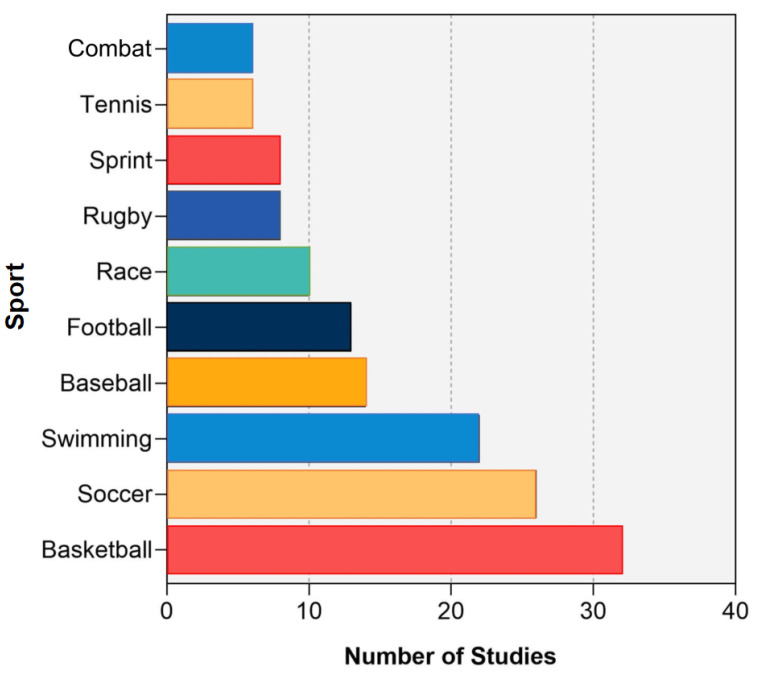
The top 10 most focused sports in adapted sport publication.

**Table 1 ijerph-19-12644-t001:** The top 10 countries of origin of papers in adapted sport research.

2001–2005	2006–2010	2011–2015	2016–2020
USA (43)	USA (50)	UK (165)	UK (178)
UK (17)	UK (38)	USA (82)	USA (171)
Australia (12)	Australia (15)	Canada (63)	Canada (85)
Canada (4)	Canada (13)	Australia (30)	Brazil (84)
Belgium (3)	China (11)	Poland (16)	Australia (65)
South Africa (3)	Germany (10)	Spain (16)	Spain (54)
France (2)	Brazil (7)	Belgium (13)	Italy (46)
Germany (2)	Netherlands (7)	China (13)	Japan (42)
Israel (2)	Italy (6)	Brazil (12)	Netherlands (37)
Netherlands (2)	France (3)	Germany (12)	South Africa (29)

**Table 2 ijerph-19-12644-t002:** The top 10 institutions contributed to publications on adapted sport research.

	Institution	Articles	Country
1.	Loughborough University	103	UK
2.	University of British Columbia	48	Canada
3.	University of Pittsburgh	45	USA
4.	University of the Sunshine Coast	45	Australia
5.	University of Groningen	42	Netherlands
6.	Catholic University of Leuven	40	Belgium
7.	University of Queensland	38	Australia
8.	German Sport University Cologne	36	Germany
9.	Harvard Medical School	36	USA
10.	Vrije University Amsterdam	35	Netherlands

**Table 3 ijerph-19-12644-t003:** The top 10 journals that published articles on adapted sport research.

Journal Title	Count
Adapted Physical Activity Quarterly	69
British Journal of Sports Medicine	51
Disability and Rehabilitation	37
International Journal of Sports Physiology and Performance	33
Sport in Society	32
Journal of Sports Sciences	30
Clinical Journal of Sport Medicine	28
Proceedings of the Institution of Civil Engineers-Civil Engineering	28
PM&R	22
Disability & Society	21

**Table 4 ijerph-19-12644-t004:** The top 10 productive authors and co-cited authors in adapted sport research.

Author	Number of Articles	Co-Cited Author	Citation
Goosey-Tolfrey V.L.	44	Webborn N.	289
Burkett B.	40	Makhov A.S.	276
Van De Vliet P.	27	Van De Vliet P.	276
Derman W.	26	Tweedy S.M.	273
Webborn N.	22	Derman W.	248
Mason B.S.	20	Howe P.D.	236
Smith B.	20	Goosey-Tolfrey V.L.	216
Reina R.	19	Stomphorst J.	207
Howe P.D.	17	Burkett B.	205
Tweedy S.M.	17	Vanlandewijck Y.C.	205

## Data Availability

Not applicable.

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
