# Peer review of "Mapping Research Trends of Adapted Sport from 2001 to 2020: A Bibliometric Analysis"

_ijerph, 2022, doi:10.3390/ijerph191912644_

Round 1

Reviewer 1 Report

 Mapping Research Trends of Adapted Sport from 2001 to 2020: A Bibliometric Analysis

The authors present a rather interesting issue of the analysis of publications in the field of adapted sport. A large range of time frames of the analyzed material is a great advantage of this publication. However, taking into account the formal publication requirements of the journal, there are significant shortcomings in the manuscript in this respect.

The basic formal defect is the citation system and the list of literature - see detailed notes, point 2.

Detailed comments:

1. The keyword Adapted sport - Line 23 was repeated

2. From the formal point of view, the manuscript does not meet the publication requirements.

 References: References must be numbered in order of appearance in the text (including table captions and figure legends) and listed individually at the end of the manuscript. We recommend preparing the references with a bibliography software package, such as EndNote, ReferenceManager or Zotero to avoid typing mistakes and duplicated references. We encourage citations to data, computer code and other citable research material. If available online, you may use reference style 9. below.

In the text, reference numbers should be placed in square brackets [], and placed before the punctuation; for example [1], [1-3] or [1,3]

3. It is not necessary to repeat the test results in the table and figure (for example: Table 1 and Figures 3 A, B or Table 3 and Figure 6).

4. Figure 7, 8, 9 use different time ranges (figure 7 no time ranges, figure 8 2001-2020, figure 9 2016-2020. Explain this.

Author Response

Based on the ratings from the reviewers, we’ve rewritten the introduction, results, and conclusion.

The reference list has also been updated.

Reviewer 1:

  1. The keyword Adapted sport - Line 23 was repeated

Response:

Deleted, thanks for the careful checking!

  1. From the formal point of view, the manuscript does not meet the publication requirements.

References: References must be numbered in order of appearance in the text (including table captions and figure legends) and listed individually at the end of the manuscript. We recommend preparing the references with a bibliography software package, such as EndNote, Reference Manager or Zotero to avoid typing mistakes and duplicated references. We encourage citations to data, computer code and other citable research material. If available online, you may use reference style 9. below.

In the text, reference numbers should be placed in square brackets [], and placed before the punctuation; for example [1], [1-3] or [1,3]

Response:

       Thanks for the clear explanations. We’ve already updated the format issues according to your suggestion.

  1. It is not necessary to repeat the test results in the table and figure (for example: Table 1 and Figures 3 A, B or Table 3 and Figure 6).

Response:

       We agree with the reviewer’s opinion. We have removed Figure 3 and the column of "Subject" in Table 3 and made revisions in the text so that no results were repeated.

  1. Figure 7, 8, 9 use different time ranges (figure 7 no time ranges, figure 8 2001-2020, figure 9 2016-2020. Explain this.

Response:

       Thanks for the careful reading and the question.

First, both Figures 7 & 8 (now Figures 6 &7) are showing the co-occurrence keywords for the past 20 years, we have added the time range for the title of Figure 6 to clarify. Figure 6 is intended to demonstrate the topic categories in different colors, while Figure 7 is used to provide information about the changes in research topics over the years.

Second, the reason to include the 2016-2020 keywords analysis is to provide a general view of the most recent topics. If it does not make sense or still sounds inappropriate, we can remove Figure 9 (now Figure 8) to make the time range more consistent.

Reviewer 2 Report

It is an interesting document on a subject not very studied and of relatively little interest in the literature of 10 years ago.

Generally, it is well written with a correct methodology, tables, and figures representing the results.

Comments

The title and purpose of the study are separated from the background. The antecedents of the introduction are insufficient and imprecise. The study is not justified and does not adequately present the problem.

What do the authors mean by the acronym TS (line 54)?

The authors mention in the text having located 11,074 documents (line 66); however, Figure 1 begins with 2,574 documents located. Where are the remaining 8,500 papers, and what was the reason for eliminating them?

Figure 1 presents wording problems.

In this figure 1, the fourth box, the authors state that they have identified 255 studies; however, according to the accounts, these were eliminated. Correct the text and mention why they were removed.

Please use the PRISMA format for the realization of this figure.

Please put as keywords words that are not found in the title or abstract. Use this critical space wisely, don't waste it.

Author Response

Overall Response:

Based on the ratings from the reviewers, we’ve rewritten the introduction, results, and conclusion.

The reference list has also been updated.

It is an interesting document on a subject not very studied and of relatively little interest in the literature of 10 years ago.

Generally, it is well written with a correct methodology, tables, and figures representing the results.

Response:

Thanks for the nice comments.

Comments

The title and purpose of the study are separated from the background. The antecedents of the introduction are insufficient and imprecise. The study is not justified and does not adequately present the problem.

Response:

       We’ve already rewritten the introduction.

       Thanks for the suggestions.

What do the authors mean by the acronym TS (line 54)?

Response:

       The TS stands for topic search. We have added the full title in the text. Thanks for the question.

The authors mention in the text having located 11,074 documents (line 66); however, Figure 1 begins with 2,574 documents located. Where are the remaining 8,500 papers, and what was the reason for eliminating them?

Figure 1 presents wording problems.

In this figure 1, the fourth box, the authors state that they have identified 255 studies; however, according to the accounts, these were eliminated. Correct the text and mention why they were removed.

Please use the PRISMA format for the realization of this figure.

Response:

       Thanks for the careful reviewing!

For the number of documents, we retrieved 2574 at the beginning and finally included 1887. The 11074 was probably an editing issue and has been updated.

And Figure 1 has been replaced based on your suggestions.

Please put as keywords words that are not found in the title or abstract. Use this critical space wisely, don't waste it.

Response:

       We appreciate the suggestion and have updated the keywords.

Round 2

Reviewer 1 Report

The manuscript was well corrected. The introduction of corrections did not change the entry in Figure 5 (line 132). Figure 6 was written, so now there are two figures 6 in the text.

Reviewer 2 Report

The authors responded appropriately to the comments.